# Clinical Impact of LAG3 Single-Nucleotide Polymorphism in DLBCL Treated with CAR-T Cell Therapy

**DOI:** 10.3390/ijms26209905

**Published:** 2025-10-11

**Authors:** Katja Seipel, Sophia Maria Spahr, Inna Shaforostova, Ulrike Bacher, Henning Nilius, Thomas Pabst

**Affiliations:** 1Department of Medical Oncology, Inselspital, Bern University Hospital, 3010 Bern, Switzerland; sophia.spahr@students.unibe.ch (S.M.S.); innaivanovna.shaforostova@insel.ch (I.S.); 2Department for Biomedical Research, University of Bern, 3008 Bern, Switzerland; 3Department of Hematology, Inselspital, Bern University Hospital, 3010 Bern, Switzerland; veraulrike.bacher@insel.ch; 4Department of Clinical Chemistry, Inselspital, Bern University Hospital, 3010 Bern, Switzerland; henning.nilius@insel.ch

**Keywords:** B-lymphocyte antigen CD19, lymphocyte activation gene 3 (LAG3), cytotoxic T-lymphocyte-associated protein 4 (CTLA-4), single nucleotide polymorphism (SNP), minor allele frequency (MAF), chimeric antigen receptor (CAR), Tisagenlecleucel (Kymriah©), Axicaptagene ciloleucel (Yescarta©), Lisocabtagene Maraleucel (Breyanzi©)

## Abstract

Lymphocyte-activation gene 3 (*LAG3*) is an immune checkpoint receptor and inhibitory regulator of T-cells. Here, we analyzed the prevalence of *LAG3* rs870849 in B-cell lymphoma patients and the treatment outcomes according to the *LAG3* genetic background and discovered that *LAG3* germline variants may affect the risk of developing lymphoma and also affect the treatment outcome of DLBCL patients in the current CD19 CAR-T cell therapies. The *LAG3* rs870849 was prevalent at high frequency in DLBCL patients. Significant differences in treatment outcomes to CAR-T cell therapy emerged in *LAG3* I455hom versus I455Thet and T455hom carriers. The overall and complete response rates to CAR-T cell therapy were lower in the I455hom genetic subgroup with median PFS in the I455hom of 2 versus 20 months in the T455hom and I455Thet subgroups (*p* = 0.025). Median OS was 6 months in the *LAG3* I455hom versus 41 months in the T455hom and I455Thet subgroups (*p* = 0.007). *LAG3* rs870849 may affect treatment outcome in CAR-T cell therapy, with favorable outcomes in T455 carriers. Specific combinations of *CTLA4* and *LAG3* germline variants may cooperate to affect the response to CAR-T cell therapy.

## 1. Introduction

Immune checkpoint regulators can modulate T-cell activation and response to CAR -T cell therapy [1,2]. CTLA-4 and LAG3 are prominent and well-studied negative regulators of activated T cells [3,4,5,6]. Novel therapeutic immune checkpoint inhibitors (ICIs) can block the negative effects of CTLA-4 and LAG3, allowing T cells to effectively target cancer cells [7]. Common germline variants of the immune checkpoint regulators can affect their expression and function, influencing susceptibility to cancer, response to immunotherapy, and the risk of immune-related adverse events [8,9,10]. In our previous study, we found that the common *CTLA4* germline variant rs231775 had clinical impact on the treatment outcomes in DLBCL patients with the current CD19 CAR-T cell therapies [11]. Lymphocyte-activation gene 3 (LAG3) protein is a promising immunotherapeutic target, with more than 20 LAG-3-targeting therapeutics in clinical trials [12]. However, in comparison to CTLA-4, LAG-3 is a multi-faceted immune receptor of great complexity [13]. The main LAG-3 ligands are Major Histocompatibility Complex class II (MHC-II) and Fibrinogen-like protein 1 (FGL1) [4]. LAG-3 can interact with the T-cell receptor (TCR) CD3 complex to inhibit TCR signal transduction, thereby terminating cell proliferation and cytokine secretion [14]. The *LAG3* genetic variant rs870849 has been associated with differences in the clinical outcome after allogeneic hematopoietic stem cell transplantation [8]. rs870849 entails the substitution of isoleucine to threonine in the LAG3 transmembrane domain (I455T). LAG3 T455 may be less effectively distributed to detergent-insoluble lipid rafts, due to the substitution of a large hydrophobic amino acid (I455) with a polar one (T455) [14]. This variation may change the kinetics of proteolytic cleavage of surface LAG3 by proteinases ADAM10 and ADAM17, thereby altering T-cell activation and cytokine production [15]. LAG-3 function involves flexible protein dimerization which influences both ligand binding and T-cell receptor (TCR) association [16,17]. Soluble LAG-3 protein (sLAG3) may compete with surface LAG-3 interactions [18,19] and has been proposed as a marker for predicting outcomes of ICI therapy [20]. The *LAG3* germline variant rs870849 arises with a minor allele frequency of 0.39 (ALFA), indicating a 15% prevalence of T455 homozygotes (T455hom), 48% I455T heterozygotes (I455Thet) and 37% I455 homozygotes (I455hom) in the European population. In this analysis, we investigated the prevalence of the *LAG3* rs870849 allele and clinical outcome in a retrospective study of a B-cell lymphoma cohort treated with current anti-CD19 CAR T-cell therapies.

## 2. Results

### 2.1. Prevalence of the LAG3 Snp rs870849 in DLBCL Patients

The sequence of the *LAG3* gene exon seven was determined in the peripheral blood cells of 112 lymphoma patients evaluated for CAR T-cell therapy at our center. A total of 25 patients (22%) carried two major alleles encoding LAG3 I455 (I455hom). A total of 60 patients (54%) had one allele with the single-nucleotide polymorphism rs870849 (I455Thet), and 27 patients (24%) carried two minor alleles with rs870849 (T455hom), indicating a minor allele frequency (MAF) of 0.49–0.53 (Figure 1). The observed MAF was higher than expected in the European population with MAF 0.39 (ALFA sample size 588090, https://www.ncbi.nlm.nih.gov/snp/, access date 22 August 2025) [21].

### 2.2. Baseline Clinical Characteristics

In total, 112 lymphoma patients who received CD19-targeted CAR-T cell therapies at Inselspital Bern were included in the study. The majority of patients were diagnosed with DLBCL (92%). Baseline clinical characteristics were analyzed for the entire cohort and for the three genetic subgroups with *LAG3* rs870849 encoding isoleucine or threonine at amino acid position 455 of the LAG-3 protein (LAG3 I455hom, I455Thet, T455hom) (Table 1). The median age at the time of initial diagnosis was 62 years in the I455hom and I455Thet, and 59 years in T455hom subgroup. The proportions of de novo and transformed DLBCL were 61% and 31%, respectively, with a higher proportion of de novo DLBCL in the I455hom subgroup (68%) and a higher proportion of transformed DLBCL in the T455hom subgroup (37%). The majority of patients presented stage four disease at the time of diagnosis according to the Ann Arbor classification system. All patients received an initial R-CHOP chemotherapy and additional chemotherapies prior to CAR-T cell therapy. Less than half of the patients received radiotherapy or hematopoietic stem cell transplantation (ASCT) with equal distribution in the three genetic subgroups.

### 2.3. Disease Features and CAR-T Cell Treatment

Clinical characteristics and details of CAR-T cell therapies varied minimally in the three genetic subgroups (Table 2). While the majority of patients in the I455hom subgroup were evaluated at IPI 3, the majority of patients in the I455Thet and T455hom subgroups were at IPI 4 (*p* = 0.31), indicating the presence of more aggressive disease with adverse prognosis. r/r DLBCL before CAR-T cell infusion presented with progressive disease in the majority of patients in all subgroups (*p* = 0.56), confirmed by PET-CT. Bridging chemotherapy was administered in 37%, bridging radiotherapy in 15% of patients, with fewer radiotherapies in the I455Thet subgroup (*p* = 0.07). All patients received lympho-depleting chemotherapy with Fludarabine and Cyclophosphamide three to five days before CAR-T cell infusion. The majority of the patients (58%) were treated with tisa-cel (Kymriah©), 37% with axi-cel (Yescarta©) and 5% with liso-cel (Breyanzi©). The proportions of CAR-T cell products varied in the three genetic subgroups with 65% tisa-cel- and 27% axi-cel-treated patients in the LAG3 I455hom versus 54% tisa-cel- and 46% axi-cel-treated patients in the LAG3 T455hom subgroup (*p* = 0.47). A total of 86 patients (76%) presented with cytokine release syndrome (CRS) of grade one (49%), grade two (24%) and grade three (4%) at a median of two days after CAR-T cell infusion. Grade two CRS was more often reached in the T455hom subgroup. A total of 37 patients (33%) presented with immune effector cell-associated neurotoxicity syndrome (ICANS) after CAR-T cell infusion at a median of six days after CAR-T cell infusion. Peak levels of inflammatory markers, C-reactive protein (CRP) and interleukin-6 (IL-6) were higher in the T455hom subgroup, in accordance with higher grade CRS. Peak levels of CAR-T cell product were detected at a median of nine days after CAR-T cell infusion with a median of 4720 copies per microgram cell-free DNA. Circulating CAR-T DNA persisted at a median of 95 copies at six months after CAR-T cell infusion.

### 2.4. Treatment Outcome Associated with LAG3 Germline Variant

Here we analyzed a cohort of 112 lymphoma patients treated with CAR-T cell therapy according to their germline *LAG3* gene variants (LAG3 I455hom, I455Thet, T455hom). The subgroups were comparable regarding baseline clinical characteristics (Table 1). The outcomes of the CAR-T cell treatments were analyzed in the entire cohort and in the three genetic subgroups (Table 3, Figure 2A,B). Patients carrying the germline variant LAG3 rs870849 had a better treatment outcome (Figure 2A,B). The overall response (OR) and complete response rates (CR) to CAR-T cell therapy were lower in the I455hom subgroup (*p* = 0.049). The median PFS in the LAG3 I455hom subgroup was 2 months versus 20 months in the T455hom and I455Thet subgroups (*p* = 0.025). The median OS in the LAG3 I455hom group was 6 months versus 41 months in the T455hom and I455Thet subgroups (*p* = 0.007). Relapse rates varied in the three genetic subgroups at 69%, 46% and 35% (*p* = 0.08) with significant differences in early relapse within three months after CAR-T cell infusion at 54%, 23% and 15%, respectively (*p* = 0.014). Death occurred in 65% of the LAG3 I455hom subgroup and 44–46% of patients with the LAG3 T455 allele (*p* = 0.18). In the multivariate analysis, the LAG3 I455hom was an indicator of inferior treatment outcome with HR 3 in PFS (*p* = 0.04) and HR 3.7 in OS (*p* = 0.008) (Table 4). Higher IPI and more prior chemotherapies were risk factors in PFS with HR 1.9 and 2.7, respectively. To address the potential bias introduced by different CAR-T product types, treatment outcomes were analyzed separately in the 42 patients with axi-cel therapy and in the 63 patients with tisa-cel therapy (Figure 2C,D). In the axi-cel-treated patients, the LAG3 I455hom subgroup had a median PFS of 11 months versus 30 months in the I455Thet and T455hom subgroups (*p* = 0.013). In the tisa-cel-treated patients, the LAG3 I455hom subgroup had a median PFS of 1 month versus over 66 months (*p* = 0.004) in the I455Thet and T455hom subgroups, and a median OS of 3 months versus over 66 months (*p* = 0.015).

### 2.5. LAG3 and CTLA4 in CAR-T Cell Response

The immune checkpoint receptors LAG-3 and CTLA-4 may cooperate to inhibit T-cell function. T-cell activation and exhaustion may differ depending on the specific combination of polymorphic protein variants of both receptors expressed in individual patients. We previously analyzed the clinical impact of *CTLA4* rs231775 in the same lymphoma cohort [11] which prompted a stratified analysis in all genetic combinations of *LAG3* rs870849 and *CTLA4* rs231775 in the current study. The outcomes of the CAR-T cell treatments were analyzed for the entire cohort and for the nine genetic combinations with group sizes of 2–27 (Table 5, Figure 3). Within the three genetic subgroups of *LAG3* I455hom, complete response rates were high in *CTLA4* A17hom, intermediate in *CTLA4* T17Ahet and low in *CTLA4* T17hom, indicating a dose-dependent effect of *CTLA4* rs231775 (Figure 3A,B). A similar pattern of response rates was exposed in the three genetic subgroups of *LAG3* I455Thet, higher in *CTLA4* A17hom, intermediate in *CTLA4* T17Ahet, and lower in *CTLA4* T17hom (Figure 3C,D), indicating a dose-dependent effect of the *CTLA4* minor allele. Within the three genetic subgroups of *LAG3* T455hom, however, the pattern of response rates differed, with high overall response rates in all *CTLA4* genetic variants (Table 5). This may indicate that the interaction of the LAG3 T455 homodimer with CTLA4 differs substantially from the interaction of the LAG3 I455 homodimer with CTLA4. Complete response rates were high (>75%) in the three genetic subgroups of *CTLA4* A17hom and in one genetic subgroup of *CTLA4* T17hom-*LAG3* T455hom. Complete response rates were low (<50%) in *LAG3* I455hom-*CTLA4* T17hom and *LAG3* T455hom-*CTLA4* T17Ahet subgroups. The genetic subgroup with the lowest survival probability included patients carrying two major alleles of both receptor genes, *LAG3* I455hom-*CTLA4* T17hom, indicating cooperative negative effects of LAG3 and CTLA4 major alleles on survival after CAR-T cell therapy. The genetic subgroups with high survival probability included patients carrying two *CTLA4* minor alleles, indicating a dominant positive effect of the *CTLA4* A17hom on survival. In patients with *LAG3* I455Thet PFS probability was high in *CTLA4* A17hom or T17Ahet and low in *CTLA4* T17hom, indicating a dominant positive effect of the *CTLA4* minor allele on PFS (Figure 3 C,D). In patients with *CTLA4* T17hom, PFS probability was intermediate in *LAG3* T455hom and low in *LAG3* T17hom, while OS probability was high in *LAG3* T455hom, intermediate in *LAG3* I455Thet, and low in *LAG3* I455hom and I455Thet (Figure 3 G,H).

## 3. Discussion

In this retrospective observational study, we analyzed treatment outcomes in B-cell lymphoma patients after CAR-T cell therapy according to the germline *LAG3* gene variants (*LAG3* I455hom, I455Thet, T455hom). The common germline variant *LAG3* rs870849 had not been previously linked to overall cancer risk. However, in our study with 112 lymphoma patients, rs870849 had a minor allele frequency of 0.49–0.53 compared to 0.39 in the European population at large (ALFA sample size 588090, https://www.ncbi.nlm.nih.gov/snp/, access date 22 August 2025), indicating an association to lymphoma risk. The median age at diagnosis was 59 years in the T455hom versus 62 years in the I455Thet and I455hom subgroups, with disease onset at an earlier age in T455hom. Moreover, the majority of patients carrying rs870849 (*LAG3* I455Thet and T455hom) had a high risk score (IPI 4) indicating aggressive disease with adverse prognosis. In contrast, the common *CTLA4* germline variant rs231775 was not associated with lymphoma risk in the same B-cell lymphoma cohort where it had a protective effect regarding disease severity [11]. In other studies, however, *CTLA4* rs231775 had been linked to lymphoma risk [22,23].

*LAG3* rs870849, like *CTLA4* rs231775, may be a prognostic marker of CAR-T cell response in DLBCL patients. In the previous study, we presented evidence for a favorable clinical outcome to CAR-T cell therapy in lymphoma patients carrying a germline *CTLA4* rs231775 [11]. In more detail, significant differences emerged in clinical outcome in *CTLA4* A17hom vs. T17Ahet and T17hom carriers with four-year progression-free survival at 77%, 59% and 30% (*p* = 0.019) and relapse rates of 20%, 37% and 56% (*p* = 0.025). This distribution indicated a dose-dependent effect of *CTLA4* rs231775, as the progression-free survival and relapse rates of the heterozygous group (T17Ahet) were in between the two homozygous groups (A17hom and T17hom). Now we present evidence of favorable clinical outcome to CAR-T cell therapy in lymphoma patients of the same cohort carrying a germline *LAG3* rs870849. In this case, the distribution of the survival rates indicated a dominant positive impact of *LAG3* rs870849. The effect of *LAG3* rs870849 was not dose-dependent as progression-free and overall survival of the LAG3 I455Thet subgroup (one minor allele) overlapped with the *LAG3* T455hom (two minor alleles) group, with inferior outcomes in *LAG3* I455hom (two major alleles). Based on experimental evidence from the literature, we propose mechanistic explanations including altered protein expression, processing, cleavage or dimerization of LAG-3 variants. LAG-3 protein dimers may form on the cell surface before proteolytic cleavage and shedding [24]. The dynamics and efficacy of LAG3 dimerization, cleavage and shedding may differ in the T455hom and I455Thet compared to the I455hom genetic backgrounds. LAG3 rs870849 may be more effectively cleaved and shed as homodimer (T455hom) and heterodimer (I455Thet) compared to the I455 homodimer which may be cleaved and shed less effectively. LAG3 cleavage and shedding is necessary for optimal T-cell function as prevention of LAG3 shedding by non-cleavable LAG3 mutants reduced T-cell function [15]. To confirm the impact of *LAG3* rs870849 on CAR-T response, and evaluate a possible role in DLBCL pathogenesis, a larger retrospective study will be required. To elucidate the molecular mechanisms, the intracellular processing, dimerization and cleavage of the two protein variants will have to be studied in vitro.

The immune checkpoint receptors LAG-3 and CTLA-4 may cooperate to inhibit T-cell function. T cell activation and exhaustion may differ depending on the specific combination of polymorphic protein variants of both receptors expressed in individual patients. We attempted to decipher the LAG3-CTLA4 interaction in a stratified analysis in all genetic combinations of *LAG3* rs870849 and *CTLA4* rs231775 in the current study. However, the combined genotypic analyses were underpowered and the results are exploratory and hypothesis-generating, rather than conclusive. Cooperative negative effects on survival after CAR-T cell therapy were inferred for patients carrying a combination of *LAG3* and *CTLA4* major alleles (LAG3 I455hom–CTLA4 T17hom). A dominant positive effect on survival was inferred for patients carrying two *CTLA4* minor alleles (*CTLA4* A17hom), independent of *LAG3* genetics. A dominant positive effect on PFS was inferred for patients carrying one *CTLA4* minor allele, and a dominant positive effect on OS for patients carrying two *LAG3* minor alleles (LAG3 T455hom), independent of *CTLA4* genetics. To confirm cooperative effects of *LAG3* rs870849 and *CTLA4* rs231775 on CAR-T response, a larger retrospective study will be required.

Based on the clinical impact of the *LAG3* genetic variant, there may be a potential benefit of LAG-3 inhibitors within the CAR-T cell therapy of DLBCL patients. If the *LAG3* rs870849 polymorphism affected the expression of the LAG-3 protein, a reduction in LAG3 function may have favorable impact on CAR-T cell treatment outcome. Several anti-LAG3 mAbs are currently in clinical trials for immunotherapy of multiple solid tumors [4]. The clinical trial RELATIVITY-022 (NCT02061761) tested the LAG3 inhibitor relatlimab in combination with the PD1 inhibitor nivolumab for the treatment or refractory B-cell malignancies [25]. A phase one trial tested the PD-1- and LAG-3-targeting bispecific molecule tebotelimab in patients with solid tumors and hematologic cancers [26]. Bavunalimab is a CTLA-4- and LAG3-targeting bispecific antibody developed to achieve anti-tumor effects by activating T cells [27]. Moreover, there are potential implications of germline variants in immune checkpoint regulators on efficacies of subsequent immunotherapies including bispecific and trispecific antibodies [28]. A future treatment strategy may be to directly delete checkpoint regulator genes on CAR-T cells to improve treatment efficacy [29].

## 4. Materials and Methods

Patients, study endpoints, statistical analysis and gene analysis were previously described [11]. Gene-specific primers covering exon seven of the LAG3 gene were forward primer 5′-CTAGTCCTGACCCTATGCCTCATCCTGT-3′ and reverse primer 5′-CCTGAGGAAGGGGTGCCATCA-3′.

## Figures and Tables

**Figure 1 ijms-26-09905-f001:**
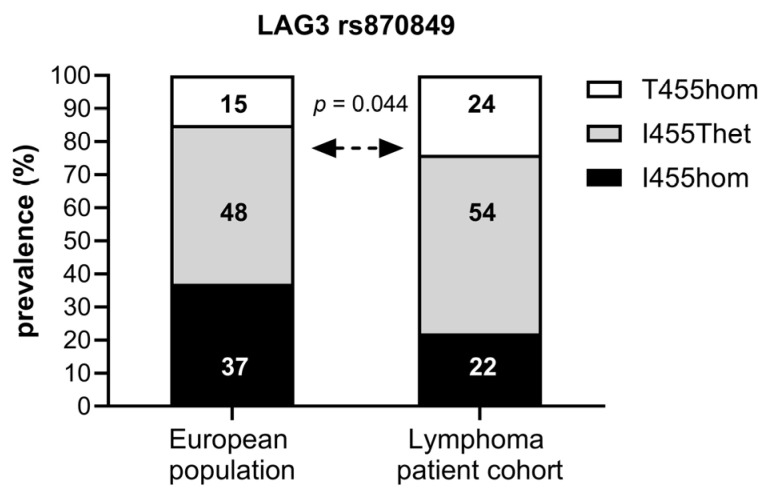
Prevalence of *LAG3* rs870849 in the European population and in the lymphoma cohort. The minor allele frequency (MAF) for rs870849 in the European population is 0.39 (ALFA) and estimated at 0.49–0.53 in the lymphoma study population.

**Figure 2 ijms-26-09905-f002:**
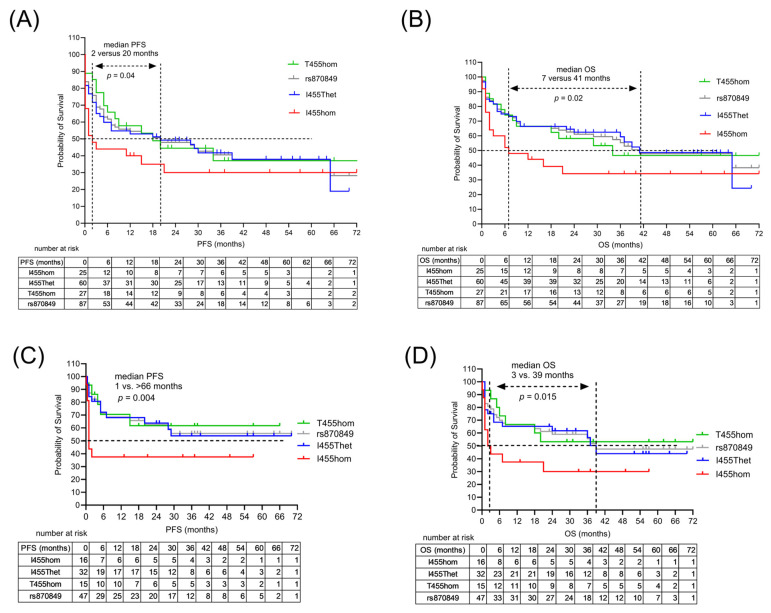
Clinical outcomes in DLBCL patients with anti-CD19 CAR-T cell therapies. Clinical outcomes in the three genetic subgroups LAG3 T455hom (green), I455Thet (blue) and I455hom (red) were analyzed in (**A**) progression-free survival (PFS) and (**B**) overall survival (OS). rs870849 (gray) reflects the combined survival of I455Thet and T455hom. Clinical outcomes in patients with tisa-cel CAR-T cell therapy in PFS (**C**) and OS (**D**).

**Figure 3 ijms-26-09905-f003:**
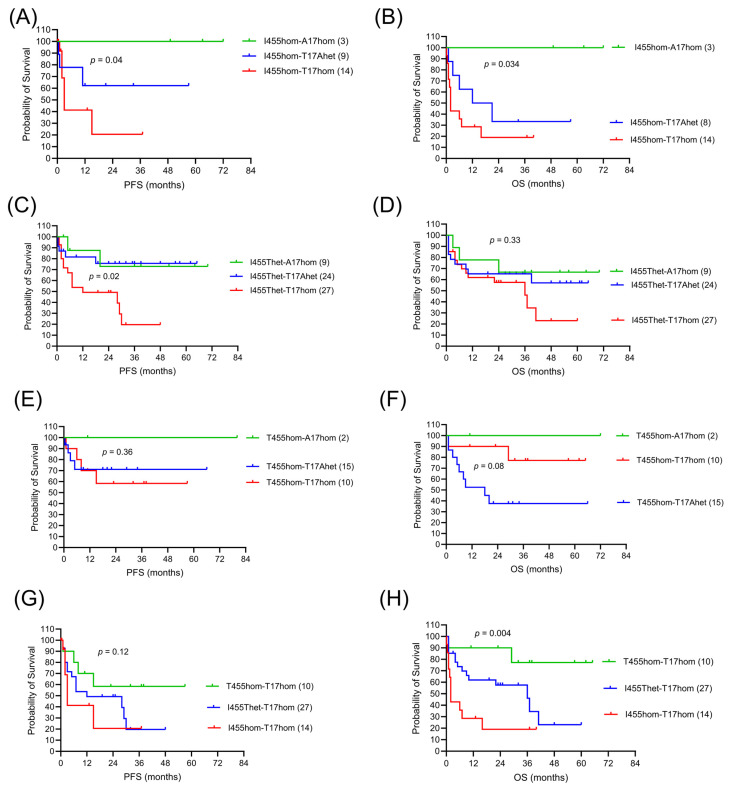
Clinical outcomes in DLBCL patients with CAR-T cell therapy according to *LAG3* and *CTLA4* genetic variants. Progression-free survival (**A**,**C**,**E**,**G**) and overall survival (**B**,**D**,**F**,**H**). Stratified survival in the *LAG3* I455hom (**A**,**B**), I455Thet (**C**,**D**), T455hom (**E**,**F**) and CTLA4 T17hom (**G**,**H**) genetic subgroups. Genetic combinations and group sizes in the different patient subgroups are indicated in the legends.

**Table 1 ijms-26-09905-t001:** Baseline clinical parameters.

Parameter	All Patients	LAG3 I455hom	LAG3 I455Thet	LAG3 T455hom	*p*-Value
Patients *n* (%)	112 (100)	25 (22)	60 (54)	27 (24)	
M:F (ratio)	64:48 (1.3)	16:9 (1.8)	32:28 (1.2)	15:12 (1.2)	0.66 ^1^
Age at ID, years median (range)	62 (25–79)	62 (36–78)	62 (29–79)	59 (25–77)	0.47 ^2^
Age at CAR-T, years median (range)	66 (26–82)	65 (37–82)	69 (30–80)	65 (26–79)	0.69 ^2^
Initial diagnosis	112 (100)				0.84 ^1^
DLBCL, *n* (%) de novo, *n* (%) transformed, *n* (%) PMBCL CLL/SLL FL, *n* (%)	103 (92) 68 (61) 35 (31) 2 (2) 1 (1) 6 (5)	24 (96) 17 (68) 8 (33) 0 0 1 (4)	53 (88) 35 (58) 18 (30) 1 (2) 1 (2) 5 (8)	26 (96) 16 (59) 10 (37) 1 (4) 0 0	
Initial stage	88 (100)	21 (24)	46 (53)	21 (23)	0.85 ^1^
One, *n* (%) Two, *n* (%) Three, *n* (%) Four, *n* (%)	5 (6) 16 (18) 18 (20) 49 (56)	0 5 (24) 5 (24) 11 (52)	3 (6) 6 (13) 9 (20) 28 (61)	2 (10) 5 (24) 4 (19) 10 (48)	
Chemotx lines	112 (100)				0.35 ^1^
1 2 3 4–6	5 (4) 41 (36) 58 (52) 7 (6)	2 (8) 5 (19) 17 (65) 1 (4)	2 (3) 24 (40) 30 (50) 4 (7)	1 (4) 12 (46) 11 (42) 2 (8)	
Radiotherapy	47 (42)	11 (48)	21 (38)	15 (55)	0.57 ^1^
Prior ASCT	41 (40)	11 (48)	19 (34)	11 (48)	0.49 ^1^

ID: initial diagnosis; CAR-T: chimeric antigen receptor T-cell therapy; DLBCL: diffuse large B-cell lymphoma; chemotx: chemotherapy; ASCT: autologous stem cell transplantation; ^1^ chi-square test; ^2^ Kruskal–Wallis test.

**Table 2 ijms-26-09905-t002:** Clinical characteristics and details of CAR-T cell treatments.

Parameter	All Patients	LAG3 I455hom	LAG3 I455Thet	LAG3 T455hom	*p*-Value
Group size, *n* (%)	112 (100)	25 (22)	60 (54)	27 (24)	
IPI					0.31 ^1^
1, *n* (%)	3 (2)	0	2 (3)	1 (4)	
2, *n* (%)	8 (7)	4 (17)	2 (3)	2 (9)	
3, *n* (%)	35 (31)	10 (40)	17 (28)	8 (30)	
4, *n* (%)	34 (31)	4 (17)	21 (35)	9 (35)	
na, *n* (%)	32 (29)	7 (27)	18 (30)	7 (27)	
Remission Status at CAR-T infusion					0.56 ^1^
CR, *n* (%)	7 (8)	2 (8)	5 (9)	0	
PR, *n* (%)	26 (29)	10 (38)	17 (32)	9 (39)	
SD, *n* (%)	4 (4)	1 (4)	3 (5)	0	
PD, *n* (%)	58 (52)	12 (48)	28 (52)	18 (67)	
Bridging chemotx	42 (37)	11 (44)	23 (43)	10 (37)	0.88 ^1^
Bridging radiotx	17 (15)	7 (30)	5 (9)	5 (22)	0.07 ^1^
LDH pre CAR-T (U/L) median (range)	337 (134–3949)	329 (145–3949)	324 (135–2355)	342 (134–1171)	0.91 ^1^
CAR-T-cell product					0.51 ^1^
Tisa-cel	64 (58)	16 (64)	33 (61)	15 (55)	
Axi-cel	42 (37)	7 (27)	23 (43)	12 (46)	
Liso-cel	6 (5)	2 (8)	4 (7)	0	
CRS	86 (76)				0.59 ^1^
Grade 1	55 (49)	15 (58)	30 (49)	10 (38)	
Grade 2	27 (24)	5 (19)	14 (23)	8 (31)	
Grade 3	4 (4)	0	2 (4)	2 (8)	
ICANS	37 (33)				0.65 ^1^
Grade 1	11 (10)	2 (9)	9 (15)	0	
Grade 2	7 (6)	2 (9)	3 (5)	2 (9)	
Grade 3	13 (11)	4 (17)	6 (10)	3 (12)	
Grade 4	6 (5)	1 (4)	4 (7)	1 (4)	
CRP peak, mg/L, median (range)	42 (2–328)	41 (3–328)	36 (2–288)	54 (4–323)	0.76 ^2^
IL-6 peak, pg/mL, median (range)	558 (4–157,117)	791 (22–49,990)	409 (4–157,117)	842 (9–8767)	0.63 ^2^
Ferritin peak, µg/L, median (range)	1323 (99–13,393)	1055 (161–13,393)	1511 (190–8690)	901 (99–4168)	0.18 ^2^
CAR-T peak (copies/µg cfDNA), median (range)	4720 (30–218,384)	4121 (320–139,656)	4641 (30–218,384)	6567 (37–35,298)	0.95 ^2^
Time to CAR-T peak, days, median (range)	9 (4–83)	8 (4–26)	9 (4–83)	9 (7–37)	0.39 ^2^
CAR-T persistence, six months, median (range)	95 (0–4061)	130 (0–2317)	74 (0–4061)	222 (0–2191)	0.18 ^2^

IPI: international prognostic index; CR: complete response; PR: partial response; SD: stable disease; PD: progressive disease; chemotx: chemotherapy; radiotx: radiotherapy; LDH: lactate dehydrogenase; cfDNA: cell-free DNA; CRS: cytokine release syndrome; ICANS: immune effector cell-associated neurotoxicity syndrome. ^1^ chi-square test; ^2^ Kruskal–Wallis test.

**Table 3 ijms-26-09905-t003:** Clinical outcome after CAR-T cell therapies in different *LAG3* germline variants.

Parameter	All Patients	LAG3 I455hom	LAG3 I455Thet	LAG3 T455hom	*p*-Value
Group size, *n* (%)	112 (100)	25 (22)	60 (54)	27 (24)	
Best response after CAR-T cell therapy					**0.049** ^1^
OR, *n* (%)	81 (72)	17 (68)	55 (90)	22 (82)	
CR, *n* (%)	56 (58)	11 (42)	43 (71)	14 (54)	
PR, *n* (%)	25 (26)	6 (24)	12 (22)	8 (27)	
SD, *n* (%)	6 (6)	4 (16)	5 (9)	3 (11)	
PD, *n* (%)	9 (9)	4 (16)	1 (4)	4 (15)	
Relapse, *n*	46 (41)	18 (69)	25 (46)	9 (35)	0.08 ^1^
Early relapse, *n*	33 (29)	14 (54)	17 (28)	4 (15)	**0.014** ^1^
Death, *n*	47 (42)	17 (65)	27 (44)	12 (46)	0.18 ^1^
PFS (mo), median	12	2	20	18	**0.025** ^2^
OS (mo), median	37	6	41	34	**0.007** ^2^

PFS: progression-free survival; OS: overall survival; CR: complete response; PR: partial response; SD: stable disease; ^1^ chi-square test; ^2^ Gehan–Breslow–Wilcoxon test on I455hom versus rs870849 (combined survival of T455hom and I455Thet). Bold formatting for *p*-values < 0.05

**Table 4 ijms-26-09905-t004:** Clinical outcome hazard ratios (HRs), multivariate analysis.

	PFS		OS	
Predictors	HR	*p*-Value	HR	*p*-Value
LAG3 I455 hom vs. T455hom	2.95	0.04	3.72	0.008
IPI 4 vs. IPI 3	1.89	0.27	1.09	0.88
Prior chemotherapy lines: 3 vs. 2	2.7	0.07	1.33	0.56
Axi-cel vs. Tisa-cel	1.35	0.58	1.37	0.55

PFS: progression-free survival, OS: overall survival; IPI: International Prognostic Index.

**Table 5 ijms-26-09905-t005:** Clinical outcome after CAR-T according to *LAG3* and *CTLA4* germline variants.

Germline Variants	Group Size	ORR	Best Response, *n* (%)			*p*-Value
*LAG3-CTLA4*	n (%)	n (%)	CR	PR	SD/PD	0.08 ^1^
I455hom-A17hom	3 (3)	3 (100)	3 (100)	0	0	
I455hom-T17Ahet	8 (7)	6 (67)	5 (62)	1 (12)	2 (25)	
I455hom-T17hom	14 (13)	8 (57)	3 (21)	5 (36)	6 (43)	
I455Thet-A17hom	9 (8)	8 (89)	7 (78)	1 (11)	1 (11)	
I455Thet-T17Ahet	24 (21)	17 (79)	16 (67)	3 (12)	5 (21)	
I455Thet-T17hom	27 (24)	22 (72)	14 (52)	8 (20)	5 (19)	
T455hom-A17hom	2 (2)	2 (100)	2 (100)	0	0	
T455hom-T17Ahet	15 (13)	13 (87)	6 (40)	7 (47)	2 (13)	
T455hom-T17hom	10 (9)	9 (90)	8 (80)	1 (10)	1 (10)	

ORR: overall response rate; CR: complete response; PR: partial response; SD: stable disease; PD: progressive disease; ^1^ chi-square test.

## Data Availability

Data available on request due to restrictions, privacy and ethics.

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
