# Peer review of "Clinical Impact of LAG3 Single-Nucleotide Polymorphism in DLBCL Treated with CAR-T Cell Therapy"

_ijms, 2025, doi:10.3390/ijms26209905_

Round 1

Reviewer 1 Report

Comments and Suggestions for Authors

The manuscript presents a retrospective analysis of germline LAG3 rs870849 variants in patients with DLBCL treated with CD19-recognizing CAR T cell therapy. The data are clearly presented, the language is precise, and the discussion integrates clinical findings with plausible biological interpretations. The study adds valuable insight into the potential prognostic role of immune checkpoint polymorphisms in the CAR T setting. However, I have some comments that could help strengthen the manuscript.

Author Response

The manuscript presents a retrospective analysis of germline LAG3 rs870849 variants in patients with DLBCL treated with CD19-recognizing CAR T cell therapy. The data is clearly presented, the language is precise, and the discussion integrates clinical findings with plausible biological interpretations. The study adds valuable insight into the potential prognostic role of immune checkpoint polymorphisms in the CAR T setting. I have several major comments that could help strengthen the manuscript: 

Response: We thank the reviewer for the thorough investigation of the manuscript. We have responded to all the comments and believe that the manuscript has been improved substantially.

Major Comments 

  • As this is a retrospective single-center study, could the authors clarify whether all consecutive CAR T-treated patients were included, or whether there was any selection (e.g., based on sample availability)? I think this would help assess potential bias.

Response: All consecutive patients were included despite differences in CAR-T cell products.

  • The authors report that the minor allele frequency (MAF) of rs870849 in their lymphoma cohort (0.45-0.48) is higher than that observed in the European reference population (0.39, ALFA). While this is an interesting observation to me, it would be important to know whether the authors have access to local control cohorts (matched for geography and sequencing method) to confirm this finding. Such a comparison could help determine whether the observed difference reflects a true predisposition signal for lymphoma rather than population or technical differences.

Response: After rechecking all sequencing data, one patient had to be reassigned from I455hom to T455hom. Accordingly, the MAF was calculated at 0.49-0.53 in the lymphoma cohort, which is a significant difference to MAF 0.39 (p = 0.044), indicating that rs870849 may be a true indicator of lymphoma risk. The ALFA sample size of the European population was 588090. Sequencing a local control cohort was not expected to reveal a MAF different from the ALFA project and could not be conducted in the time allotted for manuscript revisions.

Different CAR T products (tisagenlecleucel, axicabtagene, lisocabtagene) were used in this   study. Based on my expertise, since variable CAR T products differ in expansion kinetics and toxicity, did the authors stratify outcomes and adverse events, such as CRS severity, by product type to exclude confounding?

Response: We have added a stratified K-M analysis of tisa-cel treated patients.

  • The sample size is modest once stratified by genotype, especially in the combined LAG3-CTLA4 analyses (some groups <5 patients). Have the authors applied corrections for multiple testing or performed multivariate analyses (maybe Cox regression?) to account for known prognostic factors such as IPI, prior therapies, and product used?

Response: MVA was added in new table 4.  MVA confirms significant differences in survival.

  • I appreciate that the discussion proposes mechanistic explanations (altered cleavage/ dimerization of LAG-3 variants). While these are plausible, no functional validation is shown. Could the authors strengthen this point by citing experimental evidence from the literature or preliminary data supporting a differential biology of I455 vs. T455?

Response: We have cited experimental evidence from the literature and added a comment to the discussion on lane 239.

  • The combined genotypic analyses are very interesting but underpowered. It would be important to emphasize more explicitly in the discussion that these results are exploratory and hypothesis-generating, rather than conclusive.

Response: We have added the caveat to the discussion on lane 257.

Minor Comments

  • Line 65-66: the authors may add the terms “(T455hom)”, “(I455Thet)”, and “(I455hom)” to introduce “homozygous I455,” “heterozygous I455T,” and “homozygous T455.” At times, the terminology may be confusing to readers unfamiliar with the polymorphism.

Response: We have added the abbreviations in the introduction.

  • In Figure 3, the number of survival curves can make the interpretation difficult. I suggest splitting the analysis into separate Kaplan-Meier panels stratified by CTLA4 genotype (T/T, A/T, A/A), each showing the effect of the three LAG3 genotypes (I/I, I/T, T/T). This would improve clarity.

The authors can also consider adding new graphs as supplementary:

- K-M with the key couples of interest (e.g., the worst outcome LAG3 I/I + CTLA4 T/T vs the best outcome LAG3 T/T + CTLA4 A/A) could be highlighted in a dedicated panel to emphasize the clinical message without overwhelming the reader with too many overlapping curves.

-a forest plot of HRs (95% CI) with all 3×3 genotype combinations compared to a reference (e.g., LAG3 I/I + CTLA4 T/T; LAG3 I/I + CTLA4 A/T; LAG3 I/I + CTLA4 A/A; LAG3 I/T + CTLA4 T/T; LAG3 I/T + CTLA4 A/T; LAG3 I/T + CTLA4 A/A; LAG3 T/T + CTLA4 T/T; LAG3 T/T + CTLA4 A/T; LAG3 T/T + CTLA4 A/A).

Response: We have added the stratified KM analysis with key couples of interest.

9) Line 59-61: the authors should add one or more references.

Response. We have added one more reference.

Reviewer 2 Report

Comments and Suggestions for Authors

see attachment

Author Response

 In this manuscript the authors present data on the expression and potential role of LAG3 SNPs in the context of CAR-T cell therapy. The study adds value to the growing understanding of immune exhaustion markers and their impact on immunotherapeutic outcomes. The data are clearly presented, and the discussion is generally well-structured.

Response: We thank the reviewer for the positive comments and the valuable suggestion on the implications of LAG3 germline variants on subsequent immunotherapies.

I have some minor comments:

  1. Could the authors investigate whether there is any differential impact of common LAG3 variants on therapeutic response depending on the type of CAR-T product administered?

Response: We have done a stratified K-M analysis of tisa-cel and axi-cel treated patients. This confirmed a significant survival difference depending on LAG3 germline variants independent of CAR-T cell product. We have added the KM-analysis of the tisa-cel treated patients in Fig.2 C,D.

  1. In the discussion, the authors may also consider adding a brief comment on the potential implications of LAG3 SNPs on subsequent therapies, particularly bispecific antibodies. As these agents also rely on T-cell function and can contribute to T-cell exhaustion, it is conceivable that pre-existing germline variants of LAG3 might affect their efficacy. Highlighting this point would underscore the broader relevance of the findings in the evolving therapeutic landscape.

Response: We have added a brief comment on the potential implications of germline variants of immune checkpoint regulators on efficacies of subsequent immunotherapies in the discussion on lane 279.

Round 2

Reviewer 2 Report

Comments and Suggestions for Authors

The authors correctly addressed all the points